# Spatial mapping of b-value and fractal dimension prior to November 8, 2022 Doti Earthquake, Nepal

Ram Krishna Tiwari[1,2]*, Harihar Paudyal[2]

**1** Central Department of Physics, Tribhuvan University, Kirtipur, Kathmandu, Nepal, **2** Birendra Multiple Campus, Tribhuvan University, Bharatpur, Chitwan, Nepal

* ram.tiwari@bimc.tu.edu.np

## Abstract

An earthquake of magnitude 5.6 mb (6.6 ML) hit western Nepal (Doti region) in the wee hours of wednesday morning local time (2:12 AM, 2022.11.08) killing at least six people. Gutenberg-Richter b-value of earthquake distribution and correlation fractal dimension ($D_2$) are estimated for 493 earthquakes with magnitude of completeness 3.6 prior to this earthquake. We consider earthquakes in western Nepal Himalaya and adjoining region (80.0–83.5˚E and 27.3–30.5˚N) for the period of 1964 to 2022 for the analysis. The b-value 0.68 ±0.03 implies a high stress zone and the spatial correlation dimension 1.81±0.02 implies a highly heterogeneous region where the epicenters are spatially distributed. Low b-values and high $D_2$ values identify the study region as a high hazard zone. Focal mechanism styles and low b-values correlate with thrust nature of earthquakes and show that the earthquake's occurrence is associated with the dynamics of the faults responsible for generating the past earthquakes.

## Introduction

The magnitude distribution of earthquakes, the spatial distribution of epicenter/hypocenter, frequency of aftershocks etc., satisfy the power law distribution [1, 2]. Hence, earthquakes can be described by the scaling parameter obeying power law, known as the fractal dimension [3–5]. The b-value of frequency-magnitude is a power law involving magnitude while the two-point spatial correlation dimension of earthquake's epicenter distribution displays a power law that quantifies the proportion of randomness and clusterization. The b-value measures the material heterogeneities within a fault zone and decreases with increasing stress in the brittle part of the Earth's crust. It suggests an increasing possibility of occurrence of large magnitude earthquakes (lower b-value) and smaller magnitude earthquakes (higher b-value) [6–8]. For the regions having complex tectonic activity, b-values deviates significantly from 1.0 and in the regions characterized by stable tectonic conditions and low seismicity rates its value approaches to 1.0 over long periods of time, on the order of decades or centuries [9–11]. The b-value linked with the dynamics of individual faults is universal whereas the fractal dimension of the fault network can vary depending on geological heterogeneity [12]. Different tectonic

**Data Availability Statement:** All data are freely available from International Seismological Centre (ISC) catalog (Bondár, I., & Storchak, D. (2011). Improved location procedures at the International Seismological Centre. Geophysical Journal

International, 186(3), 1220–1244. https://doi.org/
10.1111/j.1365-246X.2011.05107.x). All figures
were created using the free available software
Python, and Generic Mapping Tools (GMT)
(Wessel et al., 2013).

**Funding:** authors received no salary or no specific
funding for this work.

**Competing interests:** no competing interest.

processes undergoing inside the earth generally activate the fault systems in asperity zones from where generous size earthquakes nucleate [13, 14].

Nepal rests on the boundary of Eurasian plate and Indian plate with major faults parallel to its length, and is vulnerable to earthquakes [15, 16]. From the northern belt to the southern belt, the South Tibetan Detachment (STD), Main Central Thrust (MCT), Main Boundary Thrust (MBT), and Main Frontal Thrust (MFT) are major faults system in the Himalaya [17, 18]. The MCT, MBT, and MFT sole into a gentle dipping detachment called the Main Himalayan Thrust (MHT) [19]. The present-day convergence between the Indian and Eurasian plates is mostly accommodated along the MHT whose surface trace is MFT. Along with these major faults, some northeast-southwest trending transverse lineaments like Tanakpur lineament, Karnali lineament, and Samea lineament are also responsible for seismic activity of the western Nepal [20]. In the past work related with anomalous seismicity of western Nepal Himalaya, potential zone for the medium size earthquakes was identified within an area bounded by 29.3˚-30.5˚N and 81.2˚-81.9˚E [21]. The recent work identified the existence of asperity capable of hosting the future earthquake with similar character of the 2015 Gorkha earthquake towards west of the epicenter of Gorkha earthquake [22]. The research carried out by separate groups [23–25] highlights the presence of a crucial seismic gap in western Nepal and adjoining region. Thus, the prime objective of this paper is to characterize the stress level and heterogeneity of the seismogenic sources by mapping b-value and fractal dimension in the region.

In record, Nepal has the long history of moderate to large earthquakes since 1255 [26]. The western region of Nepal is seismically most active segment of central Himalaya and had also hosted many earthquakes in the past and few of them are notable [20]. An earthquake of 26 September, 1964 (6.0 mb) occurred in Nepal- India border 4 km from Dharchula, Uttarakhand, India. The 6.0 mb earthquake on 27 June 1966, at the depth of 23.80 km in border of Nepal and India took the lives of 80 people [27]. The 6.1 mb Bajhang earthquakes on July 29, 1980 affected Baitadhi, Bajhang and Darchula region of western Nepal and took 125 lives [28, 29]. Following the history, a magnitude 5.6 mb or 6.6 ML earthquake hit western Nepal in the wee hours of wednesday morning local time (2:12 AM, 2022.11.08) killing at least six people [30]. The epicenter of the shock was 21 km east of Dipayal (Headquarter of Doti District), at a depth of 15.7 km. Cracks have surfaced in most of the houses in the aftermath of the event. As per the National Earthquake Monitoring & Research Center (seismonepal.gov.np), a 5.7 ML earthquake was reported in the area at 9:07 pm prior to the stronger one that was felt later. Many aftershocks had been felt after the two bigger quakes throughout the night. The jolt was felt in and around Bajhang, Kailali, Kanchanpur, Banke, Rukum west and far up to the Indian capital Delhi and lasted for about 10 seconds [31]. The fault plane solutions from the catalog of Global Centroid Moment Tensor (GCMT) [32, 33] depict the Doti earthquake as shallow angle north east dipping thrust fault event. The fault geometry of this earthquake is similar to the 1980, Bajhang earthquake (Fig 1).

## Data and methodology

For better understanding of earthquake phenomena, a decent dataset is essential. The dataset used in this study is from the catalog of International Seismological Centre (ISC) [34, 35]. We retrieved 634 earthquakes from the ISC catalog. After declustering and removing dependent events [36], only 609 earthquakes were retained. The completeness magnitude (Mc) is checked by the ZMAP software [37] which gives only 493 events with Mc $\geq$ 3.6 for final analysis (Figs 2 and 3). The final data set contains 442 events having magnitude between 3.6 and 4.9, 48 events having magnitude between 5.0 and 5.9, and three events having magnitude greater than or equal to 6.0. To map the b-value and fractal dimension, the study area was gridded at 1˚×1˚

spacing with an overlapping of 0.2°. The correlation dimension ($D_2$) and b-value are estimated only for the grids containing events $\geq 25$ for the reliable values of these parameters. The b-value is calculated by maximum likelihood estimation (MLE) method [38, 39] with standard deviation [40]. The formula used is

$$b = \frac{\log_{10} e}{M_a - \left(M_c - \frac{\Delta M}{2}\right)} \tag{1}$$

where $M_a$ is average magnitude, $M_c = 3.6$, and $\Delta M = 0.1$. The spatial correlation dimension is calculated from correlation integral function $c(r)$ [41–43] as,

$$c(r) = \frac{2}{N(N-1)} \sum_{\substack{i,j = 1 \\ i \neq j}}^{N} H(r - |X_i - X_j|) \tag{2}$$

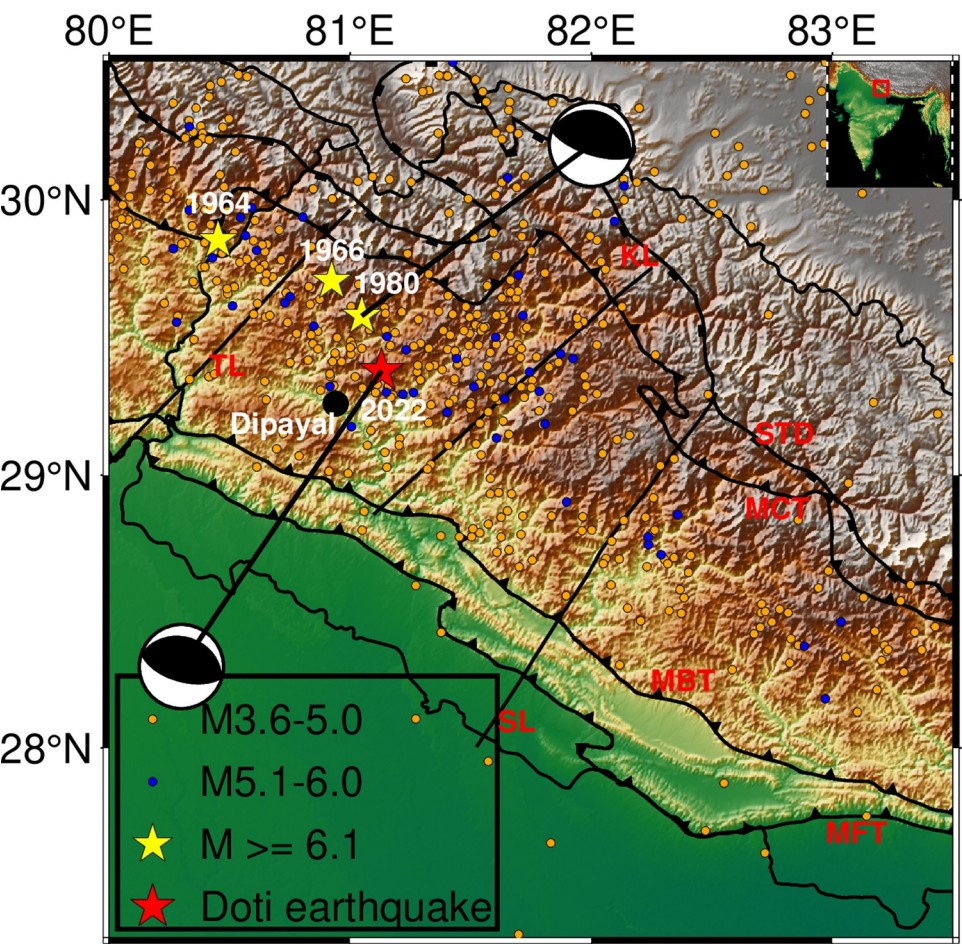

**Fig 1. Western Nepal and adjoining regions showing moderate historical earthquakes, viz.1964 earthquake, 1966 earthquake, and 1980 Bajhang earthquake (Yellow stars).** Recent Doti earthquake (2022) is represented by red star. Black sphere is indicating the Dipayal (Head quarter of Doti district). Tiny spheres (orange) indicate the earthquakes having magnitude between 3.6 and 5.0 and blue spheres indicate the earthquakes having magnitude between 5.1 and 6.0. Beach balls depict the focal mechanism solutions of July 29, 1980 Bajhang earthquake and 8 November, 2022 Doti earthquake. STD is South Tibetan Detachment, MCT is Main Central Thrust, MBT is the Main Boundary Thrust, and MFT is the Main Frontal Thrust. TL is the Tanakpur lineament, KL is Karnali lineament, and SL is Samea lineament [20].

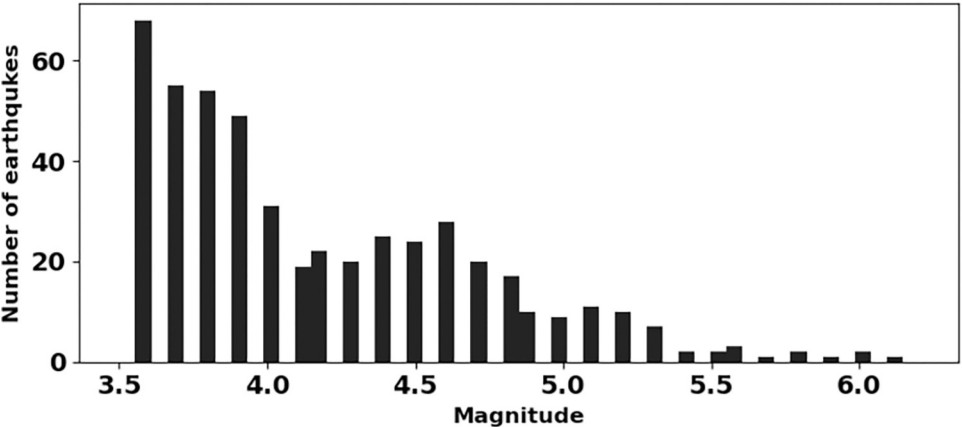

**Fig 2. Magnitude histogram of the dataset under study.**

Where r is the scaling radius (r = 5 km to r = 35 km for this study), N ranges between 25 to 171 for different grids (Table 1). The angular distance $X_i - X_j$ is evaluated by the spherical triangle method [44]. The slope of the linear part of plot for $logr$–log $c(r)$ gives the correlation fractal dimension ($D_2$) as depicted for the window containing all 493 earthquakes (Fig 4). The estimated values of $D_2$ and b are presented in the Table 1.

## Results

From the earthquake data of ISC catalog, we have estimated the values of power law exponents namely, the b-value and fractal dimension for 39 grids (sub-catalogs) and are presented in

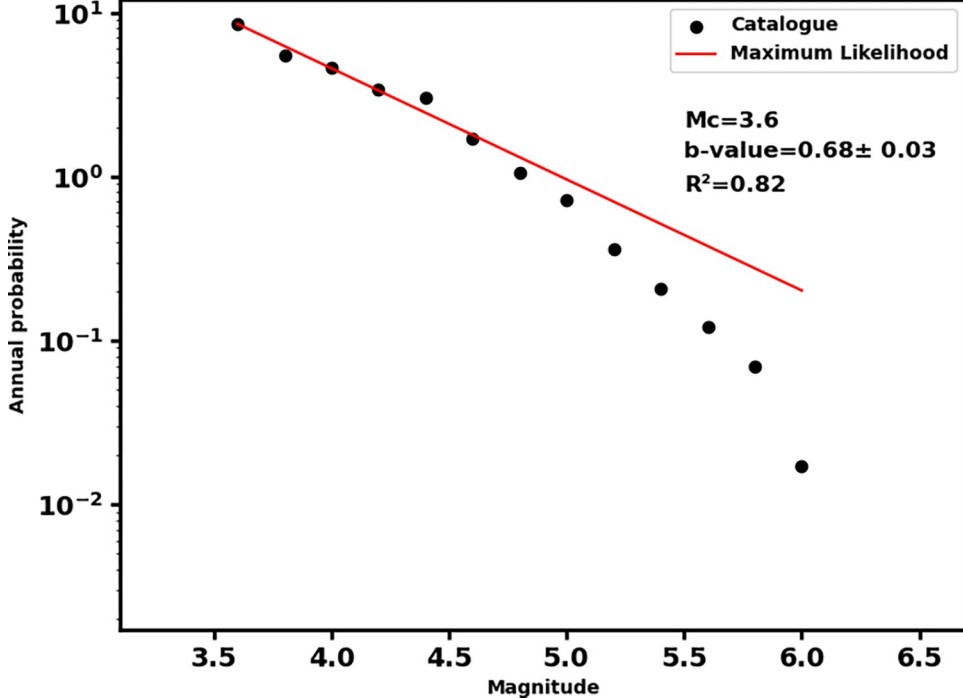

**Fig 3. Magnitude of completeness (Mc) of data set and b-value of earthquake distribution with coefficient of determination ($R^2$).**

**Table 1. List of estimated parameters of 39 grids including number of events in grid, b-value, and fractal dimension ($D_2$) with standard deviation, and respective coefficient determination ($R^2$).** Grids containing events $\geq$ 25 are only analyzed for reliable values of estimated parameters.

| S.N. | Grid | No. of events | b-value | $R^2$ | $D_2$ | $R^2$ |
|------|------|---------------|---------|-------|-------|-------|
| 1 | 80.00–81.00˚E 27.30–28.30˚N | 0 | – | – | – | – |
| 2 | 80.20–81.20˚E 27.30–28.30˚N | 0 | – | – | – | – |
| 3 | 80.40–81.40˚E 27.30–28.30˚N | 1 | – | – | – | – |
| 4 | 80.60–81.60˚E 27.30–28.30˚N | 2 | – | – | – | – |
| 5 | 80.80–81.80˚E 27.30–28.30˚N | 3 | – | – | – | – |
| 6 | 81.00–82.00˚E 27.30–28.30˚N | 4 | – | – | – | – |
| 7 | 81.20–82.20˚E 27.30–28.30˚N | 4 | – | – | – | – |
| 8 | 81.40–82.40˚E 27.30–28.30˚N | 3 | – | – | – | – |
| 9 | 81.60–82.60˚E 27.30–28.30˚N | 5 | – | – | – | – |
| 10 | 81.80–82.80˚E 27.30–28.30˚N | 5 | – | – | – | – |
| 11 | 82.00–83.00˚E 27.30–28.30˚N | 5 | – | – | – | – |
| 12 | 82.20–83.20˚E 27.30–28.30˚N | 8 | – | – | – | – |
| 13 | 82.40–83.50˚E 27.30–28.30˚N | 9 | – | – | – | – |
| 14 | 80.00–81.00˚E 28.30–29.30˚N | 18 | – | – | – | – |
| 15 | 80.20–81.20˚E 28.30–29.30˚N | 17 | – | – | – | – |
| 16 | 80.40–81.40˚E 28.30–29.30˚N | 28 | 0.57 ± 0.02 | 0.89 | 1.72 ± 0.03 | 0.99 |
| 17 | 80.60–81.60˚E 28.30–29.30˚N | 47 | 0.65 ± 0.03 | 0.86 | 1.56 ± 0.01 | 1.00 |
| 18 | 80.80–81.80˚E 28.30–29.30˚N | 62 | 0.61 ± 0.04 | 0.88 | 1.54 ± 0.03 | 0.99 |
| 19 | 81.00–82.00˚E 28.30–29.30˚N | 68 | 0.62 ± 0.04 | 0.88 | 1.61 ± 0.04 | 0.99 |
| 20 | 81.20–82.20˚E 28.30–29.30˚N | 72 | 0.66 ± 0.05 | 0.85 | 1.62 ± 0.04 | 0.99 |
| 21 | 81.40–82.40˚E 28.30–29.30˚N | 79 | 0.65 ± 0.05 | 0.94 | 1.58 ± 0.04 | 0.99 |
| 22 | 81.60–82.60˚E 28.30–29.30˚N | 63 | 0.62 ± 0.05 | 0.96 | 1.67 ± 0.03 | 0.99 |
| 23 | 81.80–82.80˚E 28.30–29.30˚N | 50 | 0.65 ± 0.06 | 0.95 | 1.55 ± 0.02 | 1.00 |

*(Continued)*

**Table 1.** (Continued)

| S.N. | Grid | No. of events | b-value | $R^2$ | $D_2$ | $R^2$ |
|---|---|---|---|---|---|---|
| 24 | 82.00–83.00˚E 28.30–29.30˚N | 49 | 0.63 ± 0.05 | 0.97 | 1.47 ± 0.02 | 1.00 |
| 25 | 82.20–83.20˚E 28.30–29.30˚N | 42 | 0.61 ± 0.03 | 0.96 | 1.38 ± 0.01 | 1.00 |
| 26 | 82.40–83.50˚E 28.30–29.30˚N | 34 | 0.69 ± 0.04 | 0.92 | 1.49 ± 0.02 | 1.00 |
| 27 | 80.00–81.00˚E 29.30–30.50˚N | 133 | 0.60 ± 0.11 | 0.88 | 1.67 ± 0.02 | 1.00 |
| 28 | 80.20–81.20˚E 29.30–30.50˚N | 147 | 0.59 ± 0.11 | 0.87 | 1.60 ± 0.03 | 0.99 |
| 29 | 80.40–81.40˚E 29.30–30.50˚N | 140 | 0.62 ± 0.11 | 0.84 | 1.64 ± 0.04 | 0.99 |
| 30 | 80.60–81.60˚E 29.30–30.50˚N | 146 | 0.64 ± 0.12 | 0.83 | 1.66 ± 0.03 | 0.99 |
| 31 | 80.80–81.80˚E 29.30–30.50˚N | 171 | 0.66 ± 0.15 | 0.82 | 1.67 ± 0.03 | 0.99 |
| 32 | 81.00–82.00˚E 29.30–30.50˚N | 159 | 0.69 ± 0.14 | 0.81 | 1.68 ± 0.03 | 1.00 |
| 33 | 81.20–82.20˚E 29.30–30.50˚N | 141 | 0.72 ± 0.15 | 0.84 | 1.64 ± 0.03 | 0.99 |
| 34 | 81.40–82.40˚E 29.30–30.50˚N | 120 | 0.70 ± 0.12 | 0.86 | 1.55 ± 0.03 | 0.99 |
| 35 | 81.60–82.60˚E 29.30–30.50˚N | 83 | 0.73 ± 0.10 | 0.81 | 1.48 ± 0.01 | 1.00 |
| 36 | 81.80–82.80˚E 29.30–30.50˚N | 40 | 0.86 ± 0.06 | 0.76 | 1.61 ± 0.02 | 1.00 |
| 37 | 82.00–83.00˚E 29.30–30.50˚N | 36 | 1.07 ± 0.08 | 0.72 | 1.72 ± 0.03 | 1.00 |
| 38 | 82.20–83.20˚E 29.30–30.50˚N | 25 | 1.40 ± 0.19 | 0.72 | 1.64 ± 0.03 | 0.99 |
| 39 | 82.40–83.50˚E 29.30–30.50˚N | 30 | 1.55 ± 0.25 | 0.67 | 1.67 ± 0.01 | 1.00 |

Table 1. The b-value describes the slope of power law and explain frequency magnitude distribution of earthquakes while fractal dimension describes the complexity of the fault system and seismogenic sources.

The b-value for whole data set is found to be 0.68 ± 0.03 while the b-value map shows the broad variation for different grids i.e., between 0.48 and 1.55 (Table 1 and Fig 3). This type of variation is acceptable for the seismically active Himalayan region [46, 47]. The broad patch of low b-value between 0.48 and 0.64 is noticed for the region occupied by historical earthquakes and the recent Doti earthquake as well (Fig 5). The low b-value areas also coincide with the major thrust system (MCT, MBT, MFT etc.) of the western Nepal.

The fractal dimension value 1.81 ± 0.02 for whole data set (Table 1 and Fig 4) and the variation between 1.36 and 1.92 for different grids indicate that the epicenters are distributed in 2D seismogenic structures. High $D_2$ contours (1.5 to 1.92) are identified for the area occupied by the past moderate earthquakes and recent Doti earthquake (Fig 6). The areas east of Dipayal are identified as the low $D_2$ contours (1.36–1.52) area. These areas could be inferred as the

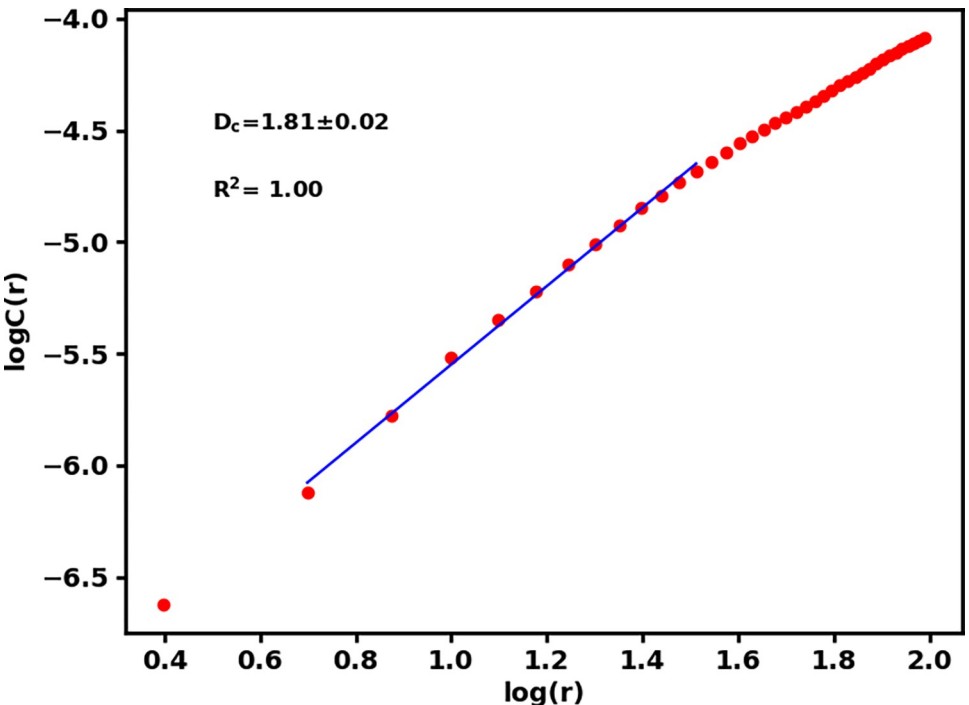

**Fig 4. The method of estimation of correlation fractal dimension as the slope of the linear part of the plot between where r = 5 km to r = 35 km is set as the distance between depopulation and saturation [45].**

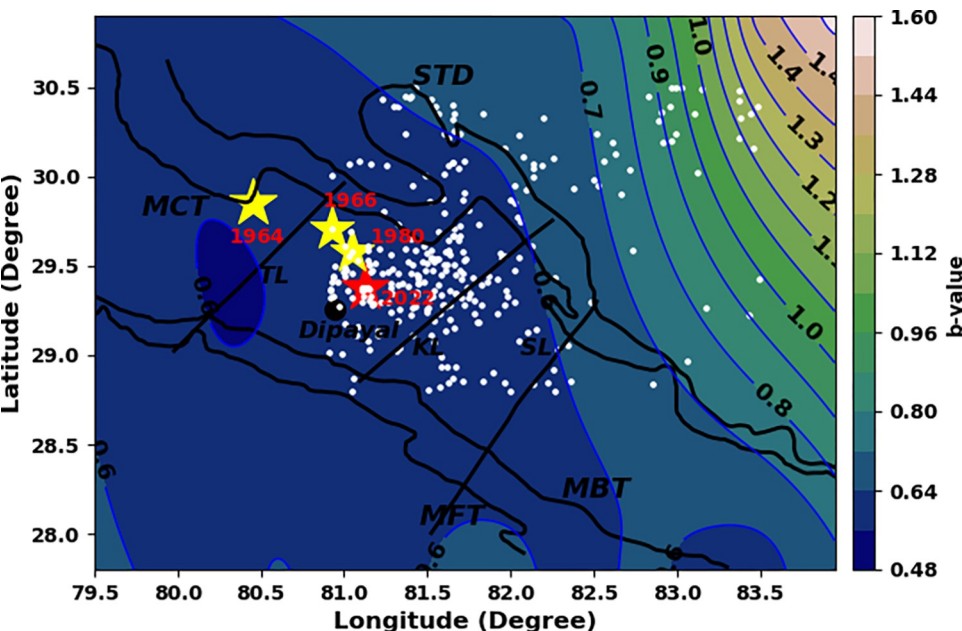

**Fig 5. b-value contours of the study region.** The b-values are calculated for the grids of size 1˚×1˚ with overlapping of 0.2˚. STD, MCT, MBT, MFT, TL, KL, and SL are as mentioned in the captions of the Fig 1. The seismic activity of the region is depicted by white dots.

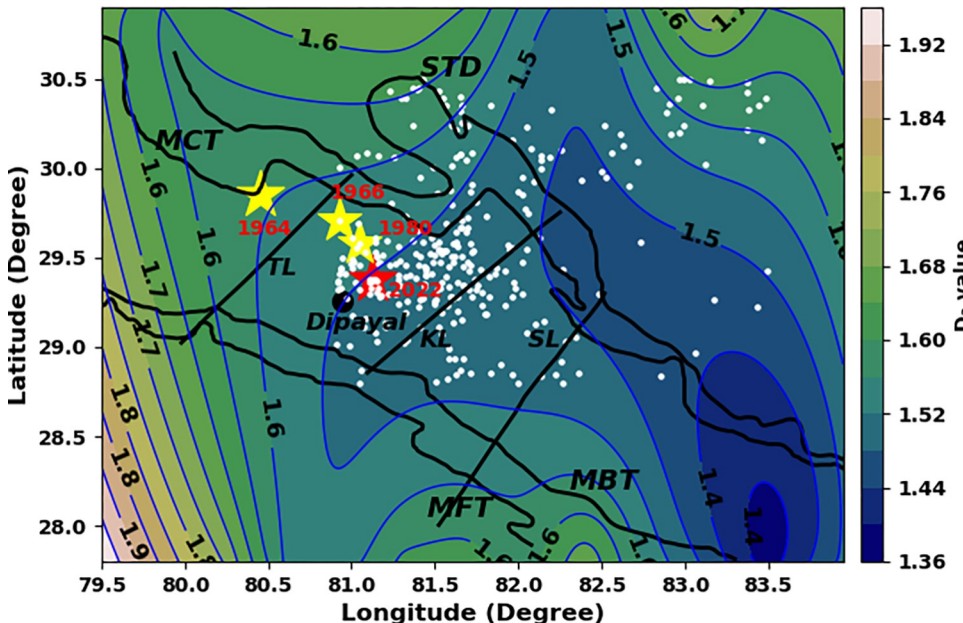

**Fig 6. D$_2$-value contours of the study region.** The D$_2$-values are calculated for the grids of size 1˚×1˚ with overlapping of 0.2˚. STD, MCT, MBT, MFT, TL, KL, and SL are as mentioned in the captions of the Fig 1. The seismic activity of the region is depicted by white dots.

asperity zones along the fault systems. The remarkably low b-value reflects that the subsurface rock mass is under stress closing to the ultimate strength, and as a result the firmly locked segment within the fault zone transforms into the state of complete failure. Moreover, high fractal dimension reflects an increase in heterogeneity of the seismogenic sources.

Fig 7 illustrates the temporal changes in b-value and D$_2$-value over a decade-long time window, with a 2-year shift between each window. To ensure reliable estimates of these parameters, only windows with 25 or more earthquakes are considered. The estimated values are presented in the Table 2. The b-value exhibits a slight increase from 1970 to 1998, followed by a sharp rise. In contrast, the D$_2$-value experiences a gradual decrease during this same time period, with a significant dip between 1988 and 1998. Subsequently, the D$_2$-value continues to decrease, with another dip observed between 2000 and 2010.

## Discussion

The low b-values estimated are the indicator of a more stressed zone or presence of asperity in the region [48, 49]. The observed low b-values can be attributed to the prevalence of dominant reverse faulting mechanisms in the area, specifically the presence of major thrusts MCT, MBT, and MFT. Thus, the area enclosed between Tanakpur Lineament and Samea lineament could be the host region of the future large earthquake. The north east corner of the study region is reflected with high b-value contours (1.00 to 1.50), so can be inferred as less probable region for the future large earthquake. Fractal dimension (D$_2$) is a measure of resistance of material against the fracture, so the fragile material had a smaller fractal dimension. The fractal dimension would increase with an increase in the energy density available for fracture [50, 51]. D$_2$ is a measure of spatial clustering and can take a value from 0 to 2. Its value close to 0 signifies that earthquakes are centralized in a small locality and the value close to 2 signifies that the earthquakes are spatially distributed [52, 53]. The higher values of fractal dimensions (between 1.36 and 1.92) obtained from this study suggest the presence of spatially distributed

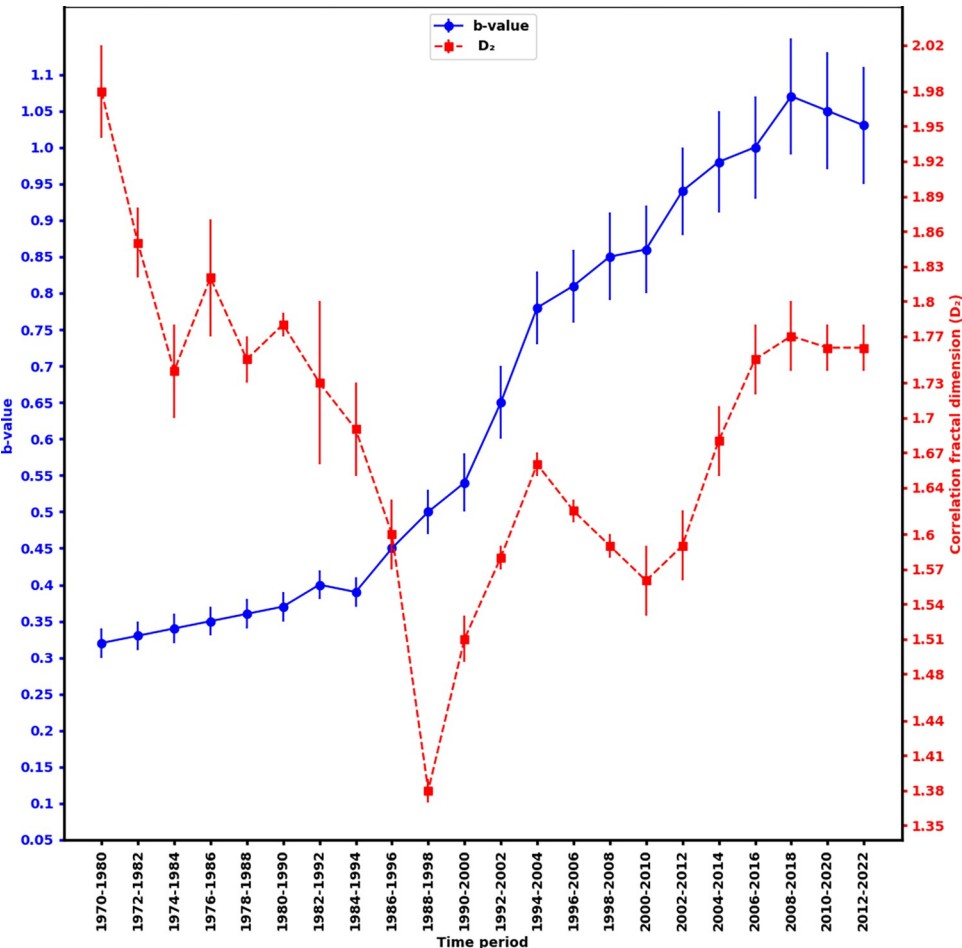

**Fig 7. The temporal changes in b-value and $D_2$-value over the period of 1970 to 2022.**

heterogeneous faults in the region and the region are seismically active. The smaller b-value zones (0.48–0.60) are corroborated with higher $D_2$ (1.5–1.9), implying negative correlation. The areas with low b-value and high $D_2$ and are expected to accumulate prominent levels of tectonic stress, which could be the risk factor for generating large earthquakes in the future [54–56]. The area identified as high risk zone in this study is within the zone identified as anomalous seismicity zone in the past work [21]. In addition, A decade wise study of b-values and $D_2$ revealed that during the initial period (1970–1994), low b-values and a gradual decrease in $D_2$ were noted, which may be related to the preparation phenomenon for the occurrence of the 1980 Bajhang earthquake. The occurrence of this earthquake would have altered the stress state of the surrounding rocks, potentially leading to the observed changes in earthquake parameters. The low b-values may indicate a gradual change in the stress field over time. A subsequent jump in b-value after the Bajhang earthquake may be due to changes in tectonic stress causing an increase in the number of small earthquakes. This increase in seismicity may have caused a temporary decrease in the fractal correlation dimension as seismic activity became more clustered around the mainshock location. As time passes and aftershocks occur, the fractal correlation dimension may increase again as the seismic activity spreads out and becomes more diffuse, leading to a more complex pattern of earthquake epicenters. Overall, these changes in the b-value and fractal correlation dimension suggest a more complex

**Table 2. List of estimated parameters of 25 windows including number of events in window, b-value, and fractal dimension ($D_2$) with standard deviation, and respective coefficient determination ($R^2$).** Windows containing events $\geq 25$ are only analyzed for reliable values of estimated parameters.

| S.N. | Window | No. of events | b-value | $R^2$ | $D_2$ | $R^2$ |
|---|---|---|---|---|---|---|
| 1 | 1964–1974 | 24 | - | – | – | – |
| 2 | 1966–1976 | 20 | – | – | – | – |
| 3 | 1968–1978 | 17 | – | – | – | – |
| 4 | 1970–1980 | 26 | 0.32 ± 0.02 | 0.92 | 1.98 ± 0.04 | 0.99 |
| 5 | 1972–1982 | 34 | 0.33 ± 0.02 | 0.91 | 1.85 ± 0.03 | 1.00 |
| 6 | 1974–1984 | 44 | 0.34 ± 0.02 | 0,91 | 1.74 ± 0.04 | 0.99 |
| 7 | 1976–1986 | 43 | 0.35 ± 0.02 | 0.90 | 1.82 ± 0.05 | 0.99 |
| 8 | 1978–1988 | 48 | 0.36 ± 0.02 | 0.88 | 1.75 ± 0.02 | 1.00 |
| 9 | 1980–1990 | 50 | 0.37 ± 0.02 | 0.87 | 1.78 ± 0.01 | 1.00 |
| 10 | 1982–1992 | 49 | 0.40 ± 0.02 | 0.89 | 1.73 ± 0.07 | 0.97 |
| 11 | 1984–1994 | 51 | 0.39 ± 0.02 | 0.90 | 1.69 ± 0.04 | 0.99 |
| 12 | 1986–1996 | 55 | 0.45 ± 0.02 | 0.92 | 1.60 ± 0.03 | 0.99 |
| 13 | 1988–1998 | 67 | 0.50 ± 0.03 | 0.94 | 1.38 ± 0.01 | 1.00 |
| 14 | 1990–2000 | 72 | 0.54 ± 0.04 | 0.92 | 1.51 ± 0.02 | 1.00 |
| 15 | 1992–2002 | 90 | 0.65 ± 0.05 | 0.90 | 1.58 ± 0.01 | 1.00 |
| 16 | 1994–2004 | 110 | 0.78 ± 0.05 | 0.83 | 1.66 ± 0.01 | 1.00 |
| 17 | 1996–2006 | 136 | 0.81 ± 0.05 | 0.81 | 1.62 ± 0.01 | 1.00 |
| 18 | 1998–2008 | 147 | 0.85 ± 0.06 | 0.81 | 1.59 ± 0.01 | 1.00 |
| 19 | 2000–2010 | 165 | 0.86 ± 0.06 | 0.81 | 1.56 ± 0.03 | 1.00 |
| 20 | 2002–2012 | 185 | 0.94 ± 0.06 | 0.77 | 1.59 ± 0.03 | 1.00 |
| 21 | 2004–2014 | 177 | 0.98 ± 0.07 | 0.75 | 1.68 ± 0.03 | 0.99 |
| 22 | 2006–2016 | 186 | 1.00 ± 0.07 | 0.73 | 1.75 ± 0.03 | 1.00 |
| 23 | 2008–2018 | 190 | 1.07 ± 0.08 | 0.70 | 1.77 ± 0.03 | 1.00 |
| 24 | 2010–2020 | 186 | 1.05 ± 0.08 | 0.71 | 1.76 ± 0.02 | 1.00 |
| 25 | 2012–2022 | 169 | 1.03 ± 0.08 | 0.73 | 1.76 ± 0.02 | 1.00 |

pattern of seismic activity in the region. The recent seismic activity observed in western Nepal can be described as a micro-fracturing process taking place within the Earth's crust prior to a major earthquake.

## Conclusion

After the analysis of earthquake dataset for the period of 58 years (1964–2022), the b-value earthquake distribution and the fractal dimension ($D_2$) of epicenter distribution are mapped for the tectonic structures of western Nepal and adjoining region. The maximum likelihood method is used for the estimation of b-value and correlation integral method is employed for the estimation of fractal dimension. The results identify the region under study as a high hazard zone with low b-values and high $D_2$ values. A high $D_2$ value obtained for the region indicates strong heterogeneity at this part of the Himalaya, may be due to varied stress level in the crust. A study conducted over multiple decades on the b-value and $D_2$ reveal the precursor signal before the Bajhang earthquake. The fault geometry of 2022 Doti earthquake and 1980 Bajhang earthquake revealed by focal mechanism solutions show the similar characteristics, so the occurrence of the earthquake could be related to the previous earthquakes. The thrust nature of focal mechanism associated with earthquakes in a region is typically the result of tectonic plate compression, which leads to a low b-value and a clustered pattern of seismicity near the fault line. This can result in a lower fractal dimension during the main shock. The mapped

region can be inferred to be most hazardous, which correlate well with seismic gap mentioned in the past literature. Finally, this study sheds new light in the understanding the characteristics of the seismogenic sources in western Nepal Himalaya.

## Data and software resources

All data are freely available from International Seismological Centre (ISC) catalog [34, 35]. All figures were created using the free available software Python, and Generic Mapping Tools (GMT) [57].

## Acknowledgments

One of the authors (RKT) would like to acknowledge both University Grant Commission (UGC), Nepal and the Tribhuvan University, Nepal for providing PhD fellowship and sabbatical leave, respectively.

## Author Contributions

**Conceptualization:** Harihar Paudyal.

**Data curation:** Ram Krishna Tiwari, Harihar Paudyal.

**Formal analysis:** Harihar Paudyal.

**Investigation:** Ram Krishna Tiwari.

**Methodology:** Ram Krishna Tiwari.

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
