## [Decision Letter · Decision Letter 0]

14 Mar 2023

PONE-D-23-00014Spatial mapping of b-value and Fractal Dimension Prior to November 8, 2022 Doti Earthquake, NepalPLOS ONE

Dear Dr. Tiwari,

Thank you for submitting your manuscript to PLOS ONE. After careful consideration, we feel that it has merit but does not fully meet PLOS ONE’s publication criteria as it currently stands. Therefore, we invite you to submit a revised version of the manuscript that addresses the points raised during the review process.

Please address all issues brought up by the reviewers with point-to-point replies. The reviewers give some suggestions for more data analysis and the content of the manuscript seems OK in general.

My main concern about the manuscript is the grammar and writing. There are a lot of grammatical errors and inconsistency in the writing style which make the manuscript difficult to follow. In the attached file "PONE-D-23-00014-AE.pdf", I listed some places where grammar can be corrected on the first page. I recommend having the manuscript proofread by a native English speaker or a language editing service.

Here are some other comments:

In Figure 2, why do the bars at 4.1, 4.8, 5.5 have no spacing?

"earth" should be capitalized as "Earth" throughout the manuscript.

We look forward to receiving your revised manuscript.

Kind regards,

Yang Li, Ph.D.

Academic Editor

PLOS ONE

“no”

“no competing interest”

7. We note that Figures 1,5 and 6 in your submission contain [map/satellite] images which may be copyrighted. All PLOS content is published under the Creative Commons Attribution License (CC BY 4.0), which means that the manuscript, images, and Supporting Information files will be freely available online, and any third party is permitted to access, download, copy, distribute, and use these materials in any way, even commercially, with proper attribution. For these reasons, we cannot publish previously copyrighted maps or satellite images created using proprietary data, such as Google software (Google Maps, Street View, and Earth). For more information, see our copyright guidelines: http://journals.plos.org/plosone/s/licenses-and-copyright.

a. You may seek permission from the original copyright holder of Figures 1,5 and 6 to publish the content specifically under the CC BY 4.0 license. 

Reviewers' comments:

Reviewer's Responses to Questions

**Comments to the Author**

1. Is the manuscript technically sound, and do the data support the conclusions?

Reviewer #1: Yes

Reviewer #2: Yes

Reviewer #3: Yes

2. Has the statistical analysis been performed appropriately and rigorously? 

Reviewer #1: Yes

Reviewer #2: Yes

Reviewer #3: Yes

3. Have the authors made all data underlying the findings in their manuscript fully available?

Reviewer #1: Yes

Reviewer #2: Yes

Reviewer #3: No

4. Is the manuscript presented in an intelligible fashion and written in standard English?

Reviewer #1: Yes

Reviewer #2: Yes

Reviewer #3: Yes

5. Review Comments to the Author

Reviewer #1: This paper focuses on calculating of Gutenberg-Richter b-value of earthquake distribution and correlation fractal dimension (D2) on the earthquake catalog in western Nepal to testify the probability of an impending earthquake. The method utilized in this paper is good. The data and the results attained in this paper can support the conclusion.

Reviewer #2: This is a solid manuscript with clear explanations about the observation, methodology and conclusions. The observation is from the standard, internationally recognized sources. The methodology is well-tested in past studies. The results and conclusions are useful for seismic hazard analysis and mitigation efforts in Nepal. I recommend publication with minor modifications.

1. For Figure 5 and 6, the authors may want to add text labels to the color bars to indicate b-value and D2.

2. The last paragraph of the introduction section is just an enumeration of past events, which is not closely related to the manuscript's main subject.

3. The equations in the data and methodology section are standard. The authors only need to give citations about them and describe how they are used in this study. Showing those equations and the explain them in detail in the manuscript dilutes the substance of the author's own work.

Reviewer #3: In this work, the authors calculated the distribution of b-values and fractal dimension of large historical earthquakes in Nepal and analyzed the implication to the Nov 8, 2022 earthquakes. Overall, I think this work is similar to a report. The methods used are standard, and the results confirm that the study region as a high hazard zone, which is not a surprise. My main concern is that the study does not provide very deep analysis and new insights about seismic statistics and earthquake risk. I only have a few minor comments:

1. How to understand the low correlation between b-value and fractal dimension in Figure 5 and 6? It will be helpful to add seismicity in these two figures too.

2. The authors briefly touch on focal mechanism. It could be helpful to add more discussion about the relationship between focal mechanism and the calculated b-value and fractal dimension.

3. I am also curious about the time-dependent b-value and fractal dimension. As the authors plot in Figure 1, there are several large earthquakes in the study region. Are the conclusion for the recent 2022 earthquake also valid for other historical earthquakes? It will be interesting to show how the b-value and fractal dimension change over time.

6. PLOS authors have the option to publish the peer review history of their article (what does this mean?). If published, this will include your full peer review and any attached files.

Reviewer #1: No

Reviewer #2: No

Reviewer #3: No

---

## [Author Response · Author response to Decision Letter 0]

16 Jun 2023

SN Reviewer’s comment Author’s Response

1. Different tectonic processes undergoing inside the earth generally activate the fault systems in asperity zones from where generous size earthquakes nucleate (Legrand, 2002; Roy et al., 2011)

Reviewer’s suggest to add period in this paragraph The period is added as decades or centuries

2. The epicenter of the shock was 21 km east of Dipayal (Headquarter of Doti District), Quake is replaced by shock

3. It is better to separate results, discussion and conclusion 

4. Please explain why two R2 One is for b-value and other for D2 value

5. Pilgrim, I., & Taylor, R. P. (2019). Fractal Analysis of Time-Series Data Sets: Now this reference is corrected in the manuscript

6. for the period should be corrected It is corrected as for the period of

7. The b-value 0.68±0.03 should not be space between 0.68 and plus minus symbol Spacing is removed from the sentence

8. A highly stress zone should be high stress zone It is written as a high stress zone

9 Identifies the study should be identify It is now corrected.

10. Brittle part of the earth’s crust Upper case E It is now corrected.

11. (Lomnitz‐ Adler, 1992). Space is deleted

12. an increase possibility of happening of corrected

13. parallel to its length, so always corrected

14. From northern belt to Corrected as the northern

15. Responsible for seismicity activity of Corrected as seismic activity

16. Magnitude greater than equal to Corrected as magnitude greater than or equal to

17. Where Ma is average magnitude, Where is corrected as where

18. In Figure 2, why do the bars at 4.1, 4.8, 5.5 have no spacing? There are earthquakes in cluster with nearly same magnitude

---

## [Decision Letter · Decision Letter 1]

25 Jul 2023

Spatial mapping of b-value and Fractal Dimension Prior to November 8, 2022 Doti Earthquake, Nepal

PONE-D-23-00014R1

Dear Dr. Tiwari,

We’re pleased to inform you that your manuscript has been judged scientifically suitable for publication and will be formally accepted for publication once it meets all outstanding technical requirements.

Kind regards,

Yang Li, Ph.D.

Academic Editor

PLOS ONE

Additional Editor Comments (optional):

Reviewers' comments:

Reviewer's Responses to Questions

**Comments to the Author**

1. If the authors have adequately addressed your comments raised in a previous round of review and you feel that this manuscript is now acceptable for publication, you may indicate that here to bypass the “Comments to the Author” section, enter your conflict of interest statement in the “Confidential to Editor” section, and submit your "Accept" recommendation.

Reviewer #1: All comments have been addressed

Reviewer #2: All comments have been addressed

2. Is the manuscript technically sound, and do the data support the conclusions?

Reviewer #1: Yes

Reviewer #2: Yes

3. Has the statistical analysis been performed appropriately and rigorously? 

Reviewer #1: Yes

Reviewer #2: Yes

4. Have the authors made all data underlying the findings in their manuscript fully available?

Reviewer #1: Yes

Reviewer #2: Yes

5. Is the manuscript presented in an intelligible fashion and written in standard English?

Reviewer #1: Yes

Reviewer #2: Yes

6. Review Comments to the Author

Reviewer #1: All comments have been responded by the authors. I am satisfied by the revision made by the Authors.

Reviewer #2: The authors have addressed my previous comments adequately.

The manuscript is technically sound and the data support the conclusion.

The statistical analysis is appropriate and rigorous.

The authors have made all data available.

The manuscript is presented in an intelligible fashion and in standard English

The authors have updated their manuscript substantially with improved readability. I recommend accept as is.

7. PLOS authors have the option to publish the peer review history of their article (what does this mean?). If published, this will include your full peer review and any attached files.

Reviewer #1: No

Reviewer #2: No

---

## [Editor Report · Acceptance letter]

31 Jul 2023

PONE-D-23-00014R1 

Spatial mapping of b-value and Fractal Dimension Prior to November 8, 2022 Doti Earthquake, Nepal 

Dear Dr. Tiwari:

I'm pleased to inform you that your manuscript has been deemed suitable for publication in PLOS ONE. Congratulations! Your manuscript is now with our production department. 

Kind regards, 

on behalf of

Dr. Yang Li 

Academic Editor

PLOS ONE